# 25-Hydroxy Vitamin D Detection Using Different Analytic Methods in Patients with Migraine

**DOI:** 10.3390/jcm8060895

**Published:** 2019-06-22

**Authors:** Luca Gallelli, Andzelika Michniewicz, Erika Cione, Aida Squillace, Manuela Colosimo, Corrado Pelaia, Alessia Fazio, Stefania Zampogna, Francesco Peltrone, Rosario Iannacchero, Giovambattista De Sarro, Monica Salerno, Giulio Di Mizio

**Affiliations:** 1Department of Health Science University of Catanzaro and Operative Unit of Clinical Pharmacology and Pharmacovigilance, Mater Domini Hospital, 88100 Catanzaro, Italy; angymich@unicz.it (A.M.); aidasquillace@libero.it (A.S.); manuelacolosimo@hotmail.it (M.C.); pelaia.corrado@gmail.com (C.P.); desarro@unicz.it (G.D.S.); 2Department of Pharmacy, Health and Nutritional Sciences, University of Calabria, 87036 Rende Cosenza, Italy; erika.cione@unical.it (E.C.); alessia.fazio@unical.it (A.F.); 3Department of Pediatry, “Pugliese-Ciaccio” Hospital, 88100 Catanzaro, Italy; stezampogna@gmail.com (S.Z.); f.peltrone@libero.it (F.P.); 4Department of Neurology, “Pugliese-Ciaccio” Hospital, 88100 Catanzaro, Italy; rosarioiann@tiscali.it; 5G&SP Working Group enclosed: Giuseppe Giuliano, Giacomo Leuzzi, Antonio Scuteri, and Francesco Corasaniti (Azienda Sanitaria Provinciale, 88100 Catanzaro, Italy), Piero Vasapollo (Azienda Sanitaria Provinciale, 88900 Crotone, Italy), Vincenzo Natale and Nazareno Brissa (Azienda Sanitaria Provinciale, 89900 Vibo Valentia, Italy); 6Forensic Medicine, Department of Medical Science, Surgical Science and advanced Technologies “G.F. Ingrassia”, University of Catania, 95124 Catania, Italy; monica.salerno@unict.it; 7Forensic Medicine, Department of Law, Economy and Sociology, Magna Graecia University of Catanzaro, 88100 Catanzaro, Italy

**Keywords:** headache, vitamin D, LC-MS/MS, EIA, HPLC-UV, CLIA, clinical risk management

## Abstract

Objectives: The aim of this study was to evaluate the performance of different analytic methods, such as liquid chromatography coupled with tandem mass spectrometry (LC-MS/MS), high-performance liquid chromatography-ultraviolet (HPLC-UV), enzyme-linked immunosorbent assay (EIA), and chemiluminescence immunoassays (CLIA), in order to highlight whether or not there is relative superiority amongst the assays. We analyzed two groups of subjects suffering from headache and two groups of healthy subjects. Design and Methods: We performed a prospective, single-blind single-center control-group study on 220 subjects with migraine. Subjects of both sexes >10 years old and with 12 months’ history of migraine were eligible for the study. As a control group, 120 healthy subjects were chosen by their family physician. Results: LC-MS/MS evaluation documented that in all enrolled subjects (migraine and control groups), the serum vitamin D3 levels were lower with respect to the normal range (30–100 ng/mL), with a mean value of 15.4 ng/mL, without difference between sex. The mean values measured using HPLC-UV, EIA, and CLIA tests such as Liaison^®^ and Architect^®^ did not show significant differences compared to the values obtained using LC-MS/MS. Conclusions: In conclusion, the population generally has low values of the vitamin D3 hormone, and the suggested range should probably be revised. HPLC-UV and CLIA were found to have appropriate analytical values compared to the reference method (LC-MS/MS), so it is possible to suggest their routine use to optimize care.

## 1. Introduction

Vitamin D is a lipophilic hormone that can be obtained from the diet and from exposure of the skin to UV in sunlight. Most foods, with the exception of fatty fish, contain little vitamin D. In some parts of the world, breakfast cereals and other products are fortified with vitamin D. Vitamin D in food exists in two forms, known as Vitamin D2 (ergocalciferol), derived from vegetables and mushrooms, and Vitamin D3 (cholecalciferol), derived from animal products. Keratinocytes produce vitamin D3 in a non-enzymatic process. The exposure of the skin to UVB (spectrum 280–320 nm) for 30–40 min per day could cover its requirements. Both Vitamin D2 and Vitamin D3 are available as oral supplements [1]. Vitamin D2 differs from D3 in having a double bond between the C22 and C23, and a methyl group in the side chain at the C24. These biochemical differences lower the affinity of D2 for the vitamin D3 binding protein (DBP), limiting its conversion into 25-hydroxyvitamin D (25(OH)D), and may result in a shorter half-life in the circulation. [2,3]. Conversion of Vitamin D3 to 25(OH)D occurs in the liver, and this form, reaching the kidney, is subsequently converted into 1,25-dihydroxy vitamin D, or calcitriol (1,25(OH)2D), by the 1α-hydroxylase. The activity of 1α-hydroxylase is upregulated by low phosphate or high parathyroid hormone (PTH) levels, and downregulated by Fibroblast-like Growth Factor-23 (FGF23). 1,25(OH)D has a half-life of 4 h and circulates in concentrations 1000-fold lower than 25(OH)D. Low levels of 1,25(OH)D may be late to develop in vitamin D3 deficiency, and because of the short half-life and low concentrations of 1,25(OH)2D, 25(OH)D is thought to be a useful marker of vitamin D3 status [4]. Hence, vitamin D3 levels are assessed by measuring the serum or plasma level of 25(OH)D [5]. Lower levels of vitamin D3 are associated with several systemic disorders involving bone, the immune system, and the musculoskeletal system, but vitamin D3 deficiency is also pointed out in subjects suffering from headache. The subjects commonly at risk for vitamin D3 deficiency include those having inadequate sun exposure, limited oral intake of it, or impaired intestinal absorption [6].

Clinical interest in the physiological importance of this steroid hormone and its possible roles in the pathophysiological processes in many diseases, spanning from metabolic to neurologic diseases, has increased the requesting for its measurement. In regard to concerns of headache, vitamin D3 insufficiency has been previously highlighted [7,8]. 

Herein, we evaluated the performance of five different techniques: Liquid chromatography coupled with tandem mass spectrometry (LC-MS/MS), high-performance liquid chromatography-ultraviolet (HPLC-UV), enzyme-linked immunosorbent assay (EIA), and chemiluminescence immunoassays (CLIA), in order to highlight the superiority amongst the assays. Besides that, the comparison of the investigated vitamin D3 levels in two different groups of subjects with migraine as well as in two groups of healthy subjects was assessed in order to evaluate the differences of vitamin D3 plasma levels in patients suffering from migraine. Furthermore, we also monitored the vitamin D3 levels after supplementation in subjects with low vitamin D3.

## 2. Methods

### 2.1. Study

We performed a prospective, single-blind single-center control-group survey from October 2017 to January 2018 in subjects admitted to the “MaterDomini” University Hospital in Catanzaro. The Local Ethics Committee approved the study protocol (protocol number 237/2017), the enrolled subjects signed the written informed consent, and the work was carried out in compliance with the Institutional Review Board/Human Subjects Research Committee requirements.

### 2.2. Samples

The study sample includes 220 subjects of both sexes enrolled in two groups. Group 1: 170 migraine subjects >18 years old (65 males and 105 females); Group 2: 50 migraine subjects <18 years old (23 males and 27 females) (Figure 1).

### 2.3. Inclusion and Exclusion Criteria

Subjects of both sexes >10 years old and with 12 months’ history of migraine, diagnosed according to the International Classification of Headache Disorders- 3rd edition criteria, were enrolled in this study.

In contrast were excluded for the study patients with severe diseases (e.g., cancer, chronic hepatitis, human immunodeficiency virus, neurodegenerative diseases), neuropsychiatric diseases (e.g., psychosis and depression, due to the possibility to reduce the compliance), kidney diseases (serum creatinine levels > 1.2 times the upper limit of the normal range), and liver disease (transaminases levels > 1.5 times the upper limit of normal ranges). Moreover, subjects with clinical conditions able to induce the development of aura (i.e., patent foramen ovale, ischemic stroke, restless legs syndrome, Parkinson’s disease, and psychiatric disorders), or with other diagnosis of headache (e.g., tension-type headache or secondary headache), or in pregnancy, and subjects who did not sign the informed consent were also excluded. Finally, subjects using medicinal preparations and dietary supplements with vitamin D3 or cod-liver oil within the last three months were not considered eligible for the study.

### 2.4. Endpoints

The first endpoint was the statistically significant differences (*p* < 0.05) in values obtained from each method of LC-MS/MS, HPLC-UV, EIA, and CLIA. The secondary endpoint was the statistically significant differences (*p* < 0.05) in migraine patients with respect to healthy subjects.

### 2.5. Experimental Protocol

#### 2.5.1. Blood Collection and Storage

Blood samples were collected from a nurse of the Laboratory of Pharmacology, in two different tubes: One screened, to avoid the effect of light, and one normal. After the collection, the samples were centrifuged, and serum aliquots were prepared and stored at −20 °C in screened and standard tubes for further analysis. Clinical data were obtained from the clinicians at the time of enrollment (blood collection), and other data (clinical characteristics of headache, co-morbidity, and drugs used) were obtained from the general practitioners of each patient. For subjects having low levels of vitamin D, a vitamin D supplement was suggested (from 10,000 UI/week to 25,000 UI/15 days), and this was bought by each patient. In these subjects, a follow-up visit was performed six months after the first diagnosis. 

#### 2.5.2. Vitamin D Detection

Levels of 25(OH)D were measured using different analytical methods with similar limits of detection (LOD), including 1-LC-MS/MS (Eureka Lab Division, Chiaravalle (Ancona) Italy; LOD 2.0 ng/mL), 2-HPLC-UV (Eureka Lab Division, Chiaravalle (Ancona) Italy; LOD 2.0 ng/mL), 3-EIA (Immunodiagnostic systems, UK; LOD 2.7 ng/mL), and 4-CLIA. This latter methodology was also assessed by three different CLIA tests: Liaison^®^ (Diasorin, Dietzenbach, Germany; LOD 2.1 ng/mL), Architect I1000^®^ (Abbott Diagnostics, Lake Forest, IL, USA; LOD 2.2 ng/mL), and Lumipulse^®^ (Fujirebio, Technologiepark 6, Belgium; LOD 2.2 ng/mL).

#### 2.5.3. LC-MS/MS Analysis

Determination of 25(OH)D was performed by a validated liquid chromatography-tandem mass spectrometry (LC-MS/MS) method. Briefly, 100 μL of human plasma were precipitated with 150 μL acetonitrile containing the internal standard, 25(OH)D. After centrifugation at 15000 rpm for 15 min, the supernatant was transferred to the vials. Analyses were performed on an AB Sciex API 4000 LC-MS/MS system (Framingham, MA, USA) fitted with an Atmospheric Pressure Chemical Ionization interface and coupled with an Agilent 1100 series LC system (Wilmington, DE, USA). Samples were analyzed using an Ascentis Express C18 column and separated employing an isocratic method with a 20:80 (*v*/*v*) water:acetonitrile mobile phase. The samples were held at 5 °C in the autosampler, and the injected volume was 10 μL at a flow rate of 0.75 mL/min. The transitions at m/z 401 → m/z 383 were monitored for 25(OH)D. Both the curtain gas and the collision gas were nitrogen, and had settings of 207 kPa (30 psi) and 34 kPa (5 psi), respectively. The ion source gas 1 was air at a setting of 448 kPa (65 psi). The needle current was set at 6 μA, and the temperature was maintained at 350 °C. The declustering potential and entrance potential were set at 70 and 10 V, respectively. The collision energy and collision exit potential were set at 16 and 26 V, respectively. The analytical performance was evaluated by the limit of detection (LOD), linearity, and average recovery for 25(OH)D, using commercial control serums. The analytical accuracy was estimated for two target concentrations: 37 and 51 ng/mL. The percent relative errors were 1.08% and 1.27%, respectively. 

#### 2.5.4. HPLC Analysis

25(OH)D analyses in serum were performed using a Shimadzu HPLC system equipped with two SCL-10-AVP pumps, an SLC-10-AVP controller, and an SPD-20A UV/vis detector. After deproteinization with a specific reagent and after purification with clean-up columns, the samples were injected directly into the HPLC system. The identification of vitamin D was performed on an RP C18 analytical column Poroshell EC 4.6 × 50 mm, 2.7 μm (Agilent, Cernusco Sul Naviglio, Milano, Italy). The analyses were conducted using a mobile phase consisting of 80% acetonitrile and 20% water in isocratic elution mode with a flow rate of 0.8 mL/ min. Injections were done using a six-port valve equipped with a 50 ul loop. The detection was carried out at 265 nm. 

#### 2.5.5. Enzyme-Linked Immunosorbent Assay (EIA)

Human sera or plasma (50 μL) were analyzed for 25(OH) vitamin D by immuno-purification followed by quantification by EIA to assess 25(OH) vitamin D levels. Briefly, samples (sera or plasma) were treated with dextran sulfate and magnesium chloride solution (to obtain a delipidated samples). These samples were then centrifuged and finally analytes were extracted using immunocapsules (50 μL sample/capsule) with monoclonal antibodies to 25(OH)D linked to solid-phase particles in suspension with vitamin D-binding protein inhibitors. Immunocapsules were agitated on a rocker–shaker for 90 min at 18–25 °C, and then washed 3 times with deionized water. After the addiction of 25(OH)D biotin solution, the samples were incubated (18–25 °C) on an orbital shaker (500–750 rpm) for 60 min and then read at 450 nm using a microplate reader (Synergy H1-BioTek) within 30 min.

#### 2.5.6. Chemiluminescence Immune Assays (CLIA)

CLIA Assay is an immunoassay technique where the label, i.e., the true “pointer” of the analytic reaction, is a luminescent molecule. Luminescence is the emission of visible or near-to-the-visible (*λ* = 300–800 nm) radiation, which is generated when an electron transitions from an excited state to the ground state. This atomic potential energy gets released in the form of light. Herein, we used Liaison^®^ (Diasorin Dietzenbach, Germany); Architect I1000^®^ (Abbott Diagnostics, Lake Forest, IL, USA); and Lumipulse^®^ (Fujirebio, Gent, Belgium). Each sample was run in triplicate.

### 2.6. Statistical Analysis

All data are expressed as mean ± standard deviation (SD). We used both nominal (sex, co-morbidity, and treatment) and categorical (age, weight, and grade of disease) variables. Moreover, we used a descriptive statistic to describe this data. Student’s *t*-test was performed to analyze the differences between each group with their control LC-MS/MS. An ANOVA test was used to evaluate the differences between the groups who either did not take or received vitamin D supplementation. Differences identified by ANOVA were examined by using an unpaired Student’s *t*-test. The threshold of statistical significance was set at * *p* < 0.05. SPSS (SPSS Inc., Chicago, IL, USA) software was used for the statistical analyses.

## 3. Results

### 3.1. Samples Collection

We enrolled 220 migraine patients, and after clinical and laboratory evaluation, 135 of these were excluded (61.36%): 98 (44,5%) because they did not meet criteria of inclusion, 10 (4.5%) because the parents did not sign the informed consent, 20 (9.1%) because they had used vitamin D before the enrollment, and 7 (3.2%) because they were not residents of Calabria and we could not obtain clinical information from their physicians (Figure 1). The remaining 85 migraine patients (38.6%) were assigned to two groups of study: Group 1: migraine > 18 years old, *n* = 45 (20 males, age 38.9 ± 5.8; 25 females, age 41 ± 7.3, *p* = 0.15); Group 2: migraine < 18 years old, *n* = 40 (20 males, age 14.0 ± 1.9; 20 females age 13.6 ± 2.1, *p* = 0.16) (Table 1). As a control group, 120 healthy subjects (90 who were >18 years old and 30 who were <18 years old) were chosen by their family doctor to match the demographics (sex, age, and co-morbidity) of the two groups of enrolled migraine patients. The clinical characteristics of the control Groups 3 and 4 (for the latter subjects who were <18 years old, parents signed the consent) are shown in Table 1.

### 3.2. Vitamin 25(OH)D Evaluation

LC-MS/MS evaluation documented that in all enrolled migraine patients (Groups 1 and 2), the serum vitamin D levels were lower with respect to the normal range of 30–100 ng/mL [9], with a mean value of 15.4 ng/mL, without differences between males and females. In migraine patients enrolled in Group 1, the vitamin D mean values were lower compared to subjects enrolled in Group 3 (14.0 ± 2.5 and 17.1 ± 5.4 ng/mL, respectively; *p* < 0.001) (Figure 2). In control subjects enrolled in Group 3 (>18 years old) and Group 4 (<18 years old), the values of Vitamin D were significantly lower with respect to the normal range (15.8 ± 2.5 and 18.3 ± 4.9 ng/mL, respectively; *p* < 0.01 with respect to normal values), but were significantly higher with respect to Groups 1 and 2 (*p* < 0.01 for Group 4 vs. Group 1; *p* < 0.05 for Group 3 vs. Group 2) (Figure 2). 

### 3.3. Vitamin 25(OH)D Differences in Methodological Approaches

The sera of enrolled subjects (Groups 1–4) were evaluated using HPLC-UV, EIA, and CLIA (Liaison^®^, Architect I1000^®^, and Lumipulse^®^). As reported in Figure 2 and Table 2, the mean values measured using HPLC-UV, Liaison^®^, and Architect^®^ did not show significant differences with values obtained using LC-MS/MS. In contrast, results obtained using the EIA test and Lumipulse^®^ were significantly different with respect to LC-MS/MS (*p* < 0.01). No statistically significant correlation was found for vitamin D concentration with the following clinical parameters: Age, Body Mass Index, activity assessment, smoke use, hypertension, diabetes mellitus, or drugs used.

### 3.4. Vitamin 25(OH)D Differences in Collection Tubes Used

In our study, we did not record any significant difference between the vitamin D values for tubes stored in the light or dark.

### 3.5. Vitamin 25(OH)D Supplementation

In all subjects enrolled in Groups 1 and 2 with vitamin D values lower than 30 ng/mL (100% if subjects), a supplementation with vitamin D was suggested (from 10,000 IU/weekly to 25,000 IU/15 days), and 6 months later, we recorded an increase in vitamin D levels (mean: 24.3 ± 3.5 ng/mL).

## 4. Discussion

In this study, we evaluated 25(OH)D plasma levels in subjects suffering from migraine using several methodological approaches. Recently, several diagnostic manufacturers have launched new 25(OH)D assays, which are aligned to the National Institute of Standards and Technology (NIST) Standard Reference Materials (SRM) (NIST, Gaithersburg, MD, USA) [9,10]. Therefore, highly automated, widely available, and less costly screening tests, with suitable accuracy, may be available to the laboratory.

Vitamin D is a lipophilic hormone involved in several homeostatic mechanisms, and lower values have been implicated in the development of several diseases [11], and lower vitamin D concentration is associated prognostically with a 35% increase in the risk of death. Some authors postulated the role of vitamin D in headache disorders, especially migraine and tension-type headache [12,13,14,15,16]. In a previous paper, we documented lower levels of 25(OH)D (13.05 ± 5.70 ng/mL) in 22 subjects with chronic migraine and/or medication-overuse headache. In this study, we documented lower levels of vitamin D in subjects with headache, irrespective of their age (less or more than 18 years old), and we also documented lower levels of 25(OH)D in control subjects (less or more than 18 years old), and therefore we are not able to suggest a specific role of this hormone in the development of headache. Previously, we documented that several drugs reduce the plasma values of 25(OH)D [17]. However, clinical evaluation excluded the role of drugs in the development of lower 25(OH)D levels. It has been reported that LC-MS/MS represents the gold standard for vitamin D plasma evaluation. Therefore, we used this approach in all subjects to evaluate if the common laboratory instruments routinely used in clinical practice (i.e., HPLC-UV, EIA, and CLIA assays) have comparable performance, and we found that the results obtained with HPLC-UV, Liaison^®^, and Architect^®^ were analytically similar to the results obtained with LC-MS/MS. Vitamin D shows several double-bonds in its chemical structure, and therefore, it can be supposed that it may be easily degraded from light exposure [18]. In particular, it has been suggested that vitamin D degradation may due to riboflavin-photosensitized oxidation by singlet oxygen, which breaks down the double-bonds of vitamin D [19]. In fact, medical laboratory sample handling guidelines currently require freezing of the sample and protecting it from artificial light, and repeated freeze–thaw cycles. Although this practice is mandatory, exposing our sample at room temperature in a clear tube under both artificial light and sunlight did not show any significant differences in 25(OH)D plasma levels with respect to the tubes screened, in agreement with another paper that documented 25(OH)D as a very stable molecule [20,21]. Taken together, these data suggest that the population has generally low values of this hormone, and the suggested range should probably be revised in agreement with recent manuscripts. In particular, Ross et al. [4] previously recommended that circulating levels of 25(OH)D above 20 ng/mL must be considered sufficient, despite a gradual acceptance over the past two decades that the lower limit of sufficiency should be 30 ng/mL. In Italy, the recommended levels of 25(OH)D are considered to be more than 30 ng/mL. The Italian Society for Osteoporosis, Mineral Metabolism, and Bone Diseases (SIOMMMS), upon revisiting the guidelines on the diagnosis, risk assessment, prevention, and management of primary and secondary osteoporosis, suggested that a normal range of vitamin D must be 30–50 ng/mL [22]. However, recently, a position statement suggested to define the cut-off for vitamin D deficiency to 20 ng/mL (50 nmol/L) in the general population and to 30 ng/mL (75 nmol/L) in subjects with bone, liver, and kidney diseases, obesity, malabsorption, pregnancy, and lactation, and the elderly [23]. In our study, the supplementation with vitamin D 10,000 IU/weekly for 6 months (oral drops) restored the serum 25(OH)D levels in all enrolled subjects. In conclusion, we failed to report a difference between migraine patients and control subjects, because all enrolled subjects had low levels of 25(OH)D. Vitamin D supplementation could restore these levels to guideline-defined sufficient concentrations. This opens a very important new scenario in terms of medical law, because specific chronic diseases or their clinical entities could probably be monitored by only evaluating the levels of 25(OH)D in the blood. In our study, HPLC-UV and the three different CLIA methods were found to have appropriate analytic values compared to the reference method (LC-MS/MS), while the ELISA assay overestimated the 25(OH)D results. LC-MS/MS is considered the gold standard for every biochemical and pharmacological evaluation, but the high cost of the equipment and the high technical expertise required are common limitations to the routine use of this method. Clinical governance requires standard analytic procedures based on validated assays that should be reliable, accurate, specific, and reproducible, in order to ensure quality improvement and patient safety. Collection of validated data as an appropriate diagnostic and therapeutic tool will lead to optimization of care by monitoring clinical activity, reducing clinical risk, and ensuring that optimum quality of care is delivered to patients thanks to reliable assays and standard operating procedures which are also effective to contain costs without negatively affecting patient care [24]. The quality of health care related to proper use of laboratory tests can be measured and verified by medical records, whose qualities are a tool to prevent disputes [25,26].

Thus, identification of a rapid, simple, reliable, and cost-effective assay, as an alternative to the reference method for the determination of 25(OH)D levels, could help in monitoring disease evolution and therapeutic strategies, improving quality of care. 

HPLC-UV or CLIA methods showed acceptable performance compared to LC-MS/MS, and are operationally easy, enable rapid testing of samples, and do not require technical expertise, so it is possible to suggest their routine use to optimize care.

In this study, we found, in contrast, that the EIA batch analysis did not provide acceptable performance compared to LC-MS/MS.

## Figures and Tables

**Figure 1 jcm-08-00895-f001:**
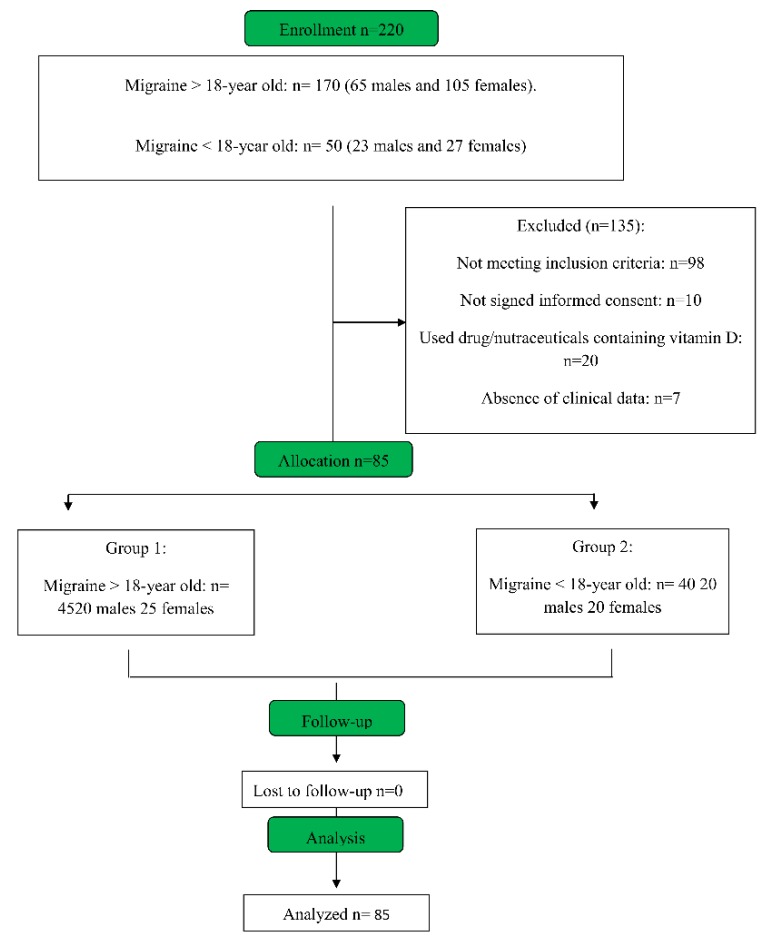
Consort diagram of migraine patients enrolled in this study.

**Figure 2 jcm-08-00895-f002:**
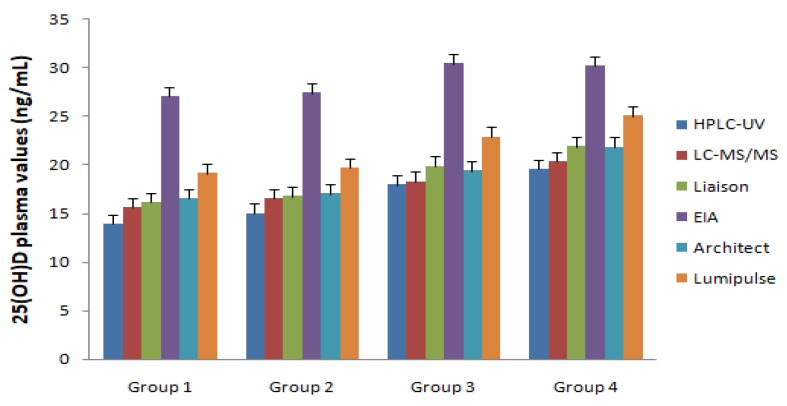
25-hydroxyvitamin D (25(OH)D) plasma values recorded in both migraine patients (Group 1: >18 years old; Group 2: <18 years old) and healthy control subjects (Group 3: >18 years old; Group 4: <18 years old) using different laboratory instruments.

**Table 1 jcm-08-00895-t001:** Demographics of migraine patients (Group 1: >18 years old; Group 2: <18 years old) and controls of healthy subjects (Group 3: >18 years old; Group 4: <18 years old) enclosed in this study. Data are expressed as mean ± standard deviation (SD).

	Group1	Group2	Group3	Group4
Male (*n*)	20	20	45	16
Female (*n*)	25	20	45	14
Age (*n*)	40 ± 6.7	13.9 ± 2	54.2 ± 8.4	13.8 ± 2.3
Body mass index	28 ± 3	24 ± 3	29 ± 6	25 ± 4
Smokers (*n*)	23	1	38	0
Blood pressure (mm/Hg)	135/88 ± 7	115/70 ± 3	143/92 ± 5	115/68 ± 4
Diabetes mellitus (*n*)	8	3	2	2
Hypertension (*n*)	29	0	33	0

**Table 2 jcm-08-00895-t002:** Vitamin 25(OH)D evaluation in enrolled subjects (Groups 1, 2, 3, and 4) using high-performance liquid chromatography-ultraviolet (HPLC-UV), enzyme-linked immunosorbent assay (EIA), and chemiluminescence immunoassay (Liaison^®^, Architect I1000^®^, and Lumipulse^®^) with respect to the gold standard liquid chromatography coupled with tandem mass spectrometry (LC-MS/MS). Data are expressed as mean ± standard deviation.

Groups	Sex	HPLC-UV (ng/mL)	LC-MS/MS (ng/mL)	Liaison (ng/mL)	EIA (ng/mL)	Architect (ng/mL)	Lumipulse (ng/mL)
Group 1	M	13.7 ± 4.1	15.4 ± 4.3	16.0 ± 5.1	27.3 ± 6.1	16.3 ± 4.0	18.7 ± 5.3
	F	14.1 ± 3.5	15.8 ± 3.7	16.3 ± 4.2	26.7 ± 6.2	16.7 ± 3.3	19.5 ± 5.5
Group 2	M	14.9 ± 4.7	16.6 ± 4.7	17.5 ± 5.9	25.8 ± 6.8	17.4 ± 4.4	20.0 ± 5.8
	F	15.1 ± 5.7	16.4 ± 6.0	16.0 ± 6.8	28.9 ± 6.2	16.6 ± 5.6	19.3 ± 6.1
Group 3	M	17.3 ± 4.3	18.2 ± 3.2	19.8 ± 3.6	30.2 ± 6.2	19.6 ± 4.1	23.6 ± 5.2
	F	18.5 ± 3.8	18.3 ± 3.6	19.9 ± 4.1	30.5 ± 5.9	19.1 ± 3.9	22.1 ± 4.8
Group 4	M	18.9 ± 4.2	19.1 ± 5.2	20.1 ± 4.8	30.7 ± 5.2	20.2 ± 4.5	23.1 ± 4.6
	F	20.2 ± 4.5	21.5 ± 4.6	23.6 ± 4.5	29.6 ± 5.1	23.4 ± 3.7	26.9 ± 5.7

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
