# Peer review of "25-Hydroxy Vitamin D Detection Using Different Analytic Methods in Patients with Migraine"

_jcm, 2019, doi:10.3390/jcm8060895_

Reviewer 1 Report

This is primarily a report dealing with six different methods of measuring levels of Vitamin D.  The methods state that two groups of headache patients were collected which included 170 that were >18 years of age and 50 subjects<18 years of age. Additionally 120 control subjects were added. There is no data on the headaches except to indicate that all 220 headache subjects had headache as defined by ICDH 3 Criteria and it is assumed the controls were headache free.  Headache is defined by the type (ICDH) and by the parameters of frequency, duration and severity. None of this data is presented and there is no attempt to correlate the headache type or characteristics with the vitamin D levels.  In addition, I was unable to determine where the data for the control subjects appeared.

This reviewer does not have the expertise to evaluate the procedures for determination of vitamin D and will leave this to other reviewers.

As the material is presented , it adds nothing to the knowledge of the relation of Headache to Vitamin D, as very little data on headache is presented. I would suggest removing the allusion to headache from the title and present this as a report dealing with different ways to measure vitamin D levels.

Author Response

Enrolled patient have migraine and this has been added in the text. Effectively, our report aimed to evaluate the different ways to measure vitamin D levels, but we evaluated this in migraine patients therefore in agreement with other comments of referee 2 and 3 we modify the title.

Reviewer 2 Report

1/ Introduction details the physiopathology (it may be shortened) but mentions in only one phrase the subjects suffering from headache. You could insist on low vitamin D described  in associations with tension-type headache , cluster headache and the supposed mechanisms (lower solar exposure with seasonal influence, dysautonomia…) and of course precise the mimicks of a single symptom complex (headache and musculoskeletal pain) 

2/ Minor spell check of English and shortening of some phrases/ideas (ex at the beginning of the introduction, beginning of discussion) 

3/ Results -> consort diagram excluded 135 but forgotten the 7 pt with lack of info? They are mentioned in the text. To be corrected in the text where is mentioned that only 98 were excluded and not 135. 

4/ Results -> group ½ in the figure 1 but groups 3 and 4? I think it will be interesting to detail this information in the figure but also in table 3 next to group 1 and 2. 

Try to emphasize the significant difference between EIA test and Lumipluse versus LC-MS/MS in the chart. 

5/ after the 6 months supplementation the vitamin D levels are still low (24.3) cf row 225  ? but mentioned as normalized (row 265) ? 

6/ You mention that eventhough all groups had a low vitamin D level, there is a significant difference between the levels of headache vs non headache (maybe it should be represented as well)  

7/ headache was diagnosed cf ICHD3, but could you detail the classification? How many patients were also having migraine, tension headache, etc and which were their treatment? 

Author Response

all enrolled patients suffered of  migraine and this has been reported in all the text and in the title also

Reviewer 3 Report

The authors performed a study on serum vitamin level D3 in a large group diagnosed as "undifferentiated" headache by using the International Classification of Headache Disorders. This a serious limit of the study. How many of them suffered from migraine, cluster headache or tension-type headache? The difference in evaluating the results in difference group make the paper  important or not.

The Authors present one previous work frothier laboratory published as Poster (Ref. 16). Please refer to a full paper if any (see ref. 4 below).

Please add and discuss previous papers (ref. 1-3) dealing with the same subject, already published but with no definitive results on the relationship of Vitamin D and headache.

1: Sohn JH, Chu MK, Park KY, Ahn HY, Cho SJ. Vitamin D deficiency in patients

with cluster headache: a preliminary study. J Headache Pain. 2018 Jul

17;19(1):54. doi: 10.1186/s10194-018-0886-7. PubMed PMID: 30019090; PubMed

Central PMCID: PMC6049846.

2: Yang Y, Zhang HL, Wu J. Is headache related with vitamin D insufficiency? J

Headache Pain. 2010 Aug;11(4):369; author reply 371. doi:

10.1007/s10194-010-0235-y. Epub 2010 Jul 3. PubMed PMID: 20602247; PubMed Central

PMCID: PMC3476355.

3: Prakash S, Mehta NC, Dabhi AS, Lakhani O, Khilari M, Shah ND. The prevalence

of headache may be related with the latitude: a possible role of Vitamin D

insufficiency? J Headache Pain. 2010 Aug;11(4):301-7. doi:

10.1007/s10194-010-0223-2. Epub 2010 May 13. Review. PubMed PMID: 20464624;

PubMed Central PMCID: PMC3476351.

4: Iannacchero R, Costa A, Squillace A, Gallelli L, Cannistrà U, De Sarro G.

P060. Vitamin D deficiency in episodic migraine, chronic migraine and

medication-overuse headache patients. J Headache Pain. 2015 Dec;16(Suppl 1):A184.

doi: 10.1186/1129-2377-16-S1-A184. PubMed PMID: 28132204; PubMed Central PMCID:

PMC4715043.

Author Response

All enrolled patients were suffered of migraine, this has been clarifying in all manuscript and also in the title

Reviewer 3

A: The authors performed a study on serum vitamin level D3 in a large group diagnosed as "undifferentiated" headache by using the International Classification of Headache Disorders. This a serious limit of the study. How many of them suffered from migraine, cluster headache or tension-type headache? The difference in evaluating the results in difference group make the paper important or not.

R: All enrolled patients were suffered of migraine, this has been clarifying in all manuscript and also in the title

A: The Authors present one previous work frothier laboratory published as Poster (Ref. 16). Please refer to a full paper if any (see ref. 4 below). Please add and discuss previous papers (ref. 1-3) dealing with the same subject, already published but with no definitive results on the relationship of Vitamin D and headache.

1: Sohn JH, Chu MK, Park KY, Ahn HY, Cho SJ. Vitamin D deficiency in patients with cluster headache: a preliminary study. J Headache Pain. 2018 Jul 17;19(1):54. doi: 10.1186/s10194-018-0886-7. PubMed PMID: 30019090; PubMed Central PMCID: PMC6049846. 2: Yang Y, Zhang HL, Wu J. Is headache related with vitamin D insufficiency? J Headache Pain. 2010 Aug;11(4):369; author reply 371. doi: 10.1007/s10194-010-0235-y. Epub 2010 Jul 3. PubMed PMID: 20602247; PubMed Central PMCID: PMC3476355. 3: Prakash S, Mehta NC, Dabhi AS, Lakhani O, Khilari M, Shah ND. The prevalence of headache may be related with the latitude: a possible role of Vitamin D insufficiency? J Headache Pain. 2010 Aug;11(4):301-7. doi: 10.1007/s10194-010-0223-2. Epub 2010 May 13. Review. PubMed PMID: 20464624; PubMed Central PMCID: PMC3476351. 4: Iannacchero R, Costa A, Squillace A, Gallelli L, Cannistrà U, De Sarro G. P060. Vitamin D deficiency in episodic migraine, chronic migraine and medication-overuse headache patients. J  headache Pain. 2015 Dec;16(Suppl 1):A184. doi: 10.1186/1129-2377-16-S1-A184. PubMed PMID: 28132204; PubMed Central PMCID: PMC4715043.

R: In agreement with your suggestion, we deleted the reference 16 and we also added the suggested references and we also discuss it in both introduction and discussion.

Round  2

Reviewer 3 Report

The manuscript has been amended correctly.